# Photoinitiated Cationic Ring-Opening Polymerization of Octamethylcyclotetrasiloxane

**DOI:** 10.3390/molecules28031299

**Published:** 2023-01-29

**Authors:** Zehra Gul Coban, Huseyin Cem Kiliclar, Yusuf Yagci

**Affiliations:** Department of Chemistry, Istanbul Technical University, 34469 Maslak, Turkey

**Keywords:** photopolymerization, onium salts, photo sensitization, cationic ring-opening polymerization, polydimethylsiloxane

## Abstract

Photochemical techniques have recently been revitalized as they can readily be adapted to different polymerization modes to yield a wide range of complex macromolecular structures. However, the implementation of the photoinduced cationic methods in the polymerization of cyclic siloxane monomers has scarcely been investigated. Octamethylcyclotetrasiloxane (D4) is an important monomer for the synthesis of polydimethylsiloxane (PDMS) and its copolymers. In this study, the cationic ring-opening polymerization (ROP) of D4, initiated by diphenyl iodonium hexafluorophosphate (DPI), has been studied. Both direct and indirect initiating systems acting at broad wavelength using benzophenone and pyrene were investigated. In both systems, photochemically generated protonic acids and silylium cations are responsible for the polymerization. The kinetics of the polymerization are followed by viscosimetry and GPC analyses. The reported approach may overcome the problems associated with conventional methods and therefore represents industrial importance for the fabrication of polysiloxanes.

## 1. Introduction

Polysiloxanes are a class of polymers with a backbone of alternating silicon and oxygen atoms with linear or branched structures. Due to their unique characteristics, such as low glass transition temperature at around −120 °C, high gas permeability, good oxidative and thermal properties, UV stability, high chain flexibility, and low surface energy, polydimethylsiloxanes (PDMS) offer critical solutions for specific needs [1,2,3,4,5]. Cyclosiloxanes, specifically octamethylcyclotetrasiloxane (D4) and hexamethylcyclotrisiloxane (D3), are the two most significant cyclic siloxane monomers for ionic ring-opening polymerization (ROP) to obtain polysiloxanes [6,7,8,9,10,11]. D4 and D3 are widely used in a broad range of applications, including automotive, medicine, personal care, implant, and adhesive industries [12,13,14].

The cationic ROP of cyclosiloxanes, initiated by strong protic acids or bases, including H_2_SO_4_, F_3_CSO_3_H and (H_3_C)_4_NOH, NaOH, and KOH, respectively, has been investigated in detail [15,16,17]. This polymerization propagates through an anionic or cationic route, depending on the type of catalyst. The problems associated with the solubility of catalysts in bulk media have negative effects on the feasibility of the method in industrial applications [18]. 

Compared to thermal polymerization methods, photopolymerization has a number of benefits, including better adaptability, higher reaction rates and spatiotemporal control, and a lower energy requirement, offering greener, safer, and more environmentally friendly reaction conditions [19,20,21,22,23]. In recent years, photoinitiated cationic ROP has been vastly studied and successfully applied to several industrially important monomers, including lactides, [24] ε-caprolactone, [25] cyclic ethers [26], and epoxides [27]. Obtained polymers in particular have deserved attention because of their influence on commercial applications, including adhesives, coatings, the encapsulation of electronic components, and 3D printing [28,29,30,31,32,33,34]. Even though there are a vast number of studies on the thermally catalyzed synthesis of PDMS, photoinduced polymerization approaches are rarely employed [35,36,37].

In this work, the photoinitiated cationic ROP of D4 by direct and indirect activation mechanisms was investigated. Benzophenone and pyrene acted as photosensitizers (PS) in the presence of diphenyliodonium (DPI) salt in indirect activation. As will be shown below, photochemically generated acids and silylium cations were electrophilic enough to induce the cationic polymerization [38,39,40].

Iodonium salts (DPI) and their photochemistry have been investigated extensively [41,42,43]. As they have significant photolysis quantum yields for the generation of reactive cationic species in the UV region, they have been widely used as a photoinitiator in cationic photopolymerizations, and to extend their spectral sensitivity to longer wavelengths, i.e., visible and NIR regions, several approaches were proposed [44,45,46,47,48,49,50,51,52]. As an oxidizing salt, the most commonly employed DPI was utilized due to its favorable redox potential to oxidize free radicals to reactive cationic species [53,54,55,56]. Such photoinduced electron transfer reactions were already reported by several research groups [57,58,59,60,61,62,63,64,65].

## 2. Results and Discussion

In the direct system, upon irradiation of DPI, the Ph-I bonds are ruptured to form iodobenzene radical cation and phenyl radical. Following this, the hydrogen abstraction of the radical cation from tetramethyldisoloxane (TMDS) essentially yields Bronsted acid capable of initiating the ROP of D4 through the protonation of oxygen on the monomer [66]. An alternative pathway, the oxidation of silyl radicals formed after hydrogen abstraction, also produces reactive silylium cations. Visible-light-induced cationic polymerization, conducted by silylium cations, was previously reported [62]. These cations are capable of initiating the ROP of cyclic monomers. The overall mechanism is presented in Figure 1. As can be seen in Table 1, an increase in the concentration of DPI and irradiation time increases conversion and molecular weight. The polymers obtained this way were characterized by FTIR, ^1^H-NMR, and ^29^Si-NMR spectral analyses (Figure 1). PDMS and D4 have structural units that only the bending of bands on the IR spectra can differentiate. FTIR spectra presented in Figure 1a indicate a strong shift around 1100 cm^−1^, which can be attributed to the ring opening of the D4 monomer. Bands belonging to other functional groups exhibit no change that underlines that polymerization occurred and no contamination of impurities present in the formed polymer. ^1^H-NMR spectra (Figure 1b) further confirm the ring-opening process by the shift of the band from 0.11 ppm to 0.08 ppm. In the 29Si-NMR spectra of the PDMS, one major peak at −22 ppm further confirms the purity of the obtained product.

After the successful implementation of the direct initiation method, sensitized approaches were taken into consideration to extend the activation to longer wavelengths. Pyrene and benzophenone were deliberately selected as PSs so as to benefit from their high wavelength absorption characteristics, high efficiency, and different protonic acid generation mechanisms. Benzophenone is known to be a triplet PS undergoing efficient hydrogen abstraction reactions. In the present case, the silyl radicals formed by the hydrogen abstraction reaction of TMDS can be oxidized to the corresponding cations to initiate the cationic ROP of D4. Concomitantly formed ketyl radicals also undergo redox reactions to essentially generate initiating protonic acids (Figure 2). Pyrene is a highly conjugated aromatic PS that forms an exciplex with DPI. Electron transfer in the exciplex generates initiating protonic acids as well as silylium cations (Figure 3).

Successful polymerizations with high yields were obtained by both benzophenone and pyrene. Control experiments were conducted to confirm that both Ph_2_I^+-^PF_6_ and PSs are indispensable for the polymerization to occur, as no polymers were formed in the absence of either component. The overall results are presented in Table 2. Notably, although a longer wavelength was employed, similar conversion and molecular weight characteristics were attained. Since the monomer and polymer are liquids, the polymerization kinetics for direct and sensitized methods were examined with the viscosity measurements. As can be seen in Figure 2 and Appendix A, the polymerization levels off after 6 h of irradiation. The observed levelling off may be due to the attack of iodobenzene produced in the system by the propagating oxonium ions. A similar observation was made in the photoinitiated cationic polymerization of cyclic ethers using iodonium salts [26]. Even though no polymerizations were observed under dark conditions in the control experiments, light on–off experiments revealed that after irradiation, the polymerization also proceeds in the dark, as no termination occurs, due to the non-nucleophilic character of the counter anion (Figure 2d).

## 3. Materials and Methods

### 3.1. Materials

Octamethylcyclotetrasiloxane (D4) (Denge Kimya, >95%), tetramethyldisiloxane (TMDS) (Denge Kimya, >95%), diphenyl iodonium hexafluorophosphate (DPI) (Aldrich, 98%), benzophenone (Merck, 99%), pyrene (Aldrich, 98%), and methanol (99.8%, Merck) were used as purchased. Dichloromethane (ISOLAB Chemicals, ≥99 %) (DCM) was purified by conventional drying and distillation methods.

### 3.2. Photoinduced Cationic ROP Procedures

#### 3.2.1. Photoinduced ROP of D4 by Direct Activation

The direct polymerization of D4 was carried out under an inert atmosphere of N_2_ using standard Schlenk techniques. D4, (0.016 mol), TMDS (1.61 × 10^−3^ mol), and the photoinitiator DPI (4.46 × 10^−3^ mol, %0.5 *w*/*w*) were placed into a sealed quartz tube. The reaction mixture was irradiated in a Rayonet merry-go-round-type photoreactor equipped with 16 Philips 8 W UV light lamps, emitting light at 300 nm with a light intensity of ~4.0 mW cm^−2^ at room temperature. The polymer was purified from the resulting reaction mixture by precipitating in methanol. A colorless, viscous liquid was obtained and washed with methanol. It was then dried for 24 h in a vacuum at ambient temperature. Approximately similar conversions and molecular weight distributions were obtained with replicates. The highest numbers were reported. The exact reaction conditions were repeated for a viscosimetric analysis. The measurements were performed by using collected aliquot samples at specified times.

#### 3.2.2. Photoinduced ROP of D4 by Sensitization

The photosensitized polymerization of D4 was carried out under an inert atmosphere of N_2_ using standard Schlenk techniques. D4, (0.016 mol), TMDS (1.61 × 10^−3^ mol), DPI (4.46 × 10^−3^ mol, %0.5 *w*/*w*), and the corresponding PS (4.46 × 10^−3^ mol) were placed into a sealed quartz tube. Polymerizations were performed using a Rayonet merry-go-round-type photoreactor equipped with 16 Philips 8 W/O6 lamps, emitting light nominally at 360 nm. The light intensity was ~3.0 mW cm^−2^, as measured by the Delta Ohm model HD-9021 radiometer. Approximately similar conversions and molecular weight distributions were obtained with replicates. The highest numbers were reported. The exact reaction conditions were repeated for a viscosimetric analysis. The measurements were performed by using collected aliquot samples at specified times.

### 3.3. Instrumentation

^1^H-NMR spectra were recorded at room temperature at 500 MHz (128 scans per analysis) on an Agilent VNMRS 500 spectrometer in CDCl_3_, using tetramethyl silane as an internal standard. ^29^Si-NMR spectra were recorded at room temperature using glass tubes at 500 MHz (256 scans; relax time of 30 s) on an Agilent VNMRS 500 spectrometer in CDCl_3_, without tetramethyl silane as an internal standard. Gel permeation chromatography (GPC) measurements were performed on a TOSOH EcoSEC GPC system equipped with an auto sampler system, a temperature-controlled pump, a column oven, a refractive index (RI) detector, a purge and degasser unit, and a TSKgel superhZ2000, 4.6 mm ID × 15 cm × 2 cm column. Tetrahydrofuran was used as an eluent at a flow rate of 1.0 mL.min^−1^ at 40 °C. The refractive index detector was calibrated with polystyrene standards, having narrow molecular weight distributions. The data were analyzed using Eco-SEC analysis software. The Fourier transform infrared (FTIR) spectroscopy measurements were recorded on a PerkinElmer FTIR Spectrum One spectrometer. Viscosimetry experiments were conducted at room temperature via Ubbelohde Apparatus SI Analytics Typ 501 01/0a Appr. Nr. 1068985. 5.

## 4. Conclusions

The photoinitiated polymerization of D4 using DPI salt under ambient temperature has been demonstrated. Both direct and indirect photosensitization approaches were effectively adapted in the polymerization process. Both approaches involved the photogeneration of Bronsted acids and/or silylium cations to initiate the cationic ROP of D4. The polymer structure has been proven by ^1^H-NMR and FTIR analyses. Kinetic investigations indicated that polymerization reaches a steady level after certain irradiation time. A future practical benefit could be foreseen in connection with the fact that the presented methodology, based on the possibility of photoinduced synthesis of polysiloxanes at desired wavelengths by using common compounds under environmentally friendly and lower energy conditions, has the possibility of conveying its technological aspect on an industrial level. Future studies will be devoted to the photoinduced synthesis of polysiloxanes copolymers in connection with the specialty application areas.

## Data Availability

The data set presented in this study is available in this article.

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
