# Peer review of "Photoinitiated Cationic Ring-Opening Polymerization of Octamethylcyclotetrasiloxane"

_molecules, 2023, doi:10.3390/molecules28031299_

Round 1

Reviewer 1 Report

The manuscript “Photoinitiated cationic ring-opening polymerization of octamethylcyclotetrasiloxane” by Yagci and coworkers reports on the possible mechanisms of D4 polymerization using photochemical methods. The advancement of photochemical methods is of interest in the field of silicone science and worthy of exploration, particularly when high Mw polymers can be produced using environmentally benign methods.

While the manuscript is easy to understand, and written generally, in grammatically correct English, with only a few minor issues noted, there are several larger issues that must be addressed before this manuscript can be accepted for publication. 

1.     There is no discussion of silyl cations, and the requirements needed to form such intermediates, in the introduction. Furthermore, there is little discussion of ROP of cyclic siloxanes (cationic or anionic) included in the introduction. This is surprising given the large amount of literature available on the subject. A single reference from 1955 is not sufficient to cover this volume of work available. Including some background here would set the stage for the current work. 

2.     It does not appear that any control reactions were performed. Are all components required in each reaction. A statement addressing should be included in the main text.

3. There is no indication that any of the reactions were performed more than once. The number of replicates should be stated in the Materials and Methods section.

4.     There is no evidence provided to justify the formation of a silyl cation as an intermediate. Given the extreme electrohilicilty of silyl cations I question this statement.

5.     In a paper purporting to discuss the polymerization of cyclosiloxanes I am expecting to see some 29Si NMR characterization, none is provided.

6.     The authors do not clearly state which of their two possible mechanisms they believe is responsible: “silyl cation” formation or initiation by hexafluorophosphoric acid. Provide experimental evidence and make a definitive statement to support either, or both, mechanisms.

7.     In Figure 2, the fitted lines do not really fit the data that well, particularly in panel 2a. This is misleading to the reader. Error bars need to be included on all data points. 

8. 30% of all the included citations are to the author’s own work. This is too many.

Other minor issues that need to be addressed.

Line 29-30 - A single reference from 1955 for a statement claiming that D3 and D4 are the two most important monomers for ionic ROP is not appropriate. This needs to be expanded upon, and better referenced.

Line 52/53 - sufficiently electrophiles is confusing.

Line 55 - If iodinium salts have been extensively investigated, more than a single reference is required.

Line 58-59 - Most of these references are to the authors own work.

Line 61 – Change raptured to ruptured.

Line 110 – Where was isopropyl alcohol used? This is the only mention of this chemical in the manuscript.

Line 117 and 127 - What is the volume of the reaction? Concentration of reactants? Number of replicates? The synthetic methods need more detail.

Line 134 – NMR solvent? What internal standard was used? How many scans?

Line 158 - Are there supplementary materials available? 

Author Response

Response to Reviewer 1 Comments

General Statement: The manuscript “Photoinitiated cationic ring-opening polymerization of octamethylcyclotetrasiloxane” by Yagci and coworkers reports on the possible mechanisms of D4 polymerization using photochemical methods. The advancement of photochemical methods is of interest in the field of silicone science and worthy of exploration, particularly when high Mw polymers can be produced using environmentally benign methods.

While the manuscript is easy to understand, and written generally, in grammatically correct English, with only a few minor issues noted, there are several larger issues that must be addressed before this manuscript can be accepted for publication.

General Response: We thank the Reviewer 1 for her/his kind words and for finding the work interesting and worthy to be explored.

Point 1: There is no discussion of silyl cations, and the requirements needed to form such intermediates, in the introduction. Furthermore, there is little discussion of ROP of cyclic siloxanes (cationic or anionic) included in the introduction. This is surprising given the large amount of literature available on the subject. A single reference from 1955 is not sufficient to cover this volume of work available. Including some background here would set the stage for the current work.  

Response 1: Discussions on silyl cations and detailed literature research have now been included in the main text as advised.

Point 2: It does not appear that any control reactions were performed. Are all components required in each reaction. A statement addressing should be included in the main text.

Response 2: Control experiments were already done. A proper explanation has been now included in the main text.

Point 3: There is no indication that any of the reactions were performed more than once. The number of replicates should be stated in the Materials and Methods section.

Response 3: Now the detailed explanation has been included.

Point 4: There is no evidence provided to justify the formation of a silyl cation as an intermediate. Given the extreme electrohilicilty of silyl cations I question this statement.

Response 4: The formation of silyl cations previously was proven by using acylsilane compounds as photoinitiators. The related work has been addressed in the revised manuscript. In the presnt work, no spectral differences are observable on the obtained products since silyl cations and HPF6 superacid yields similar polymers without any significant difference.

Point 5: In a paper purporting to discuss the polymerization of cyclosiloxanes I am expecting to see some 29Si NMR characterization, none is provided.

Response 5: 29Si-NMR spectra have now been provided as suggested. It was not previously included as our instrument is not capable to analyze 29Si nuclei and no crucial differences were expected between the monomer and polymer.

Point 6: The authors do not clearly state which of their two possible mechanisms they believe is responsible: “silyl cation” formation or initiation by hexafluorophosphoric acid. Provide experimental evidence and make a definitive statement to support either, or both, mechanisms.

Response 6: After the initiation of the cationic ROP, no spectral difference is observable between the chains initiated by HPF6  or generated silyl cations. Though, it is known that following the protonation of the monomer, ring opening occurs and thus formed silyl cations are electrophile enough to propagate the polymerization. Additional information on silylium cations regarding cationic photopolymerization done by acyl silanes is now mentioned in the main text.

Point 7: In Figure 2, the fitted lines do not really fit the data that well, particularly in panel 2a. This is misleading to the reader. Error bars need to be included on all data points.

Response 7: Details of the mathematical modeling and the obtained results are now included in SI. Different line fitting approaches have been done and the figures amended accordingly.

Point 8: 30% of all the included citations are to the author’s own work. This is too many.

Response 8: Selfcitations were unintentional since our research group is one of the leading groups in the related field. Now several references have been included in the paper to reduce the percentage.

Minor Points: Other minor issues that need to be addressed.

Line 29-30 - A single reference from 1955 for a statement claiming that D3 and D4 are the two most important monomers for ionic ROP is not appropriate. This needs to be expanded upon, and better referenced.

Line 52/53 - sufficiently electrophiles is confusing.

Line 55 - If iodinium salts have been extensively investigated, more than a single reference is required.

Line 58-59 - Most of these references are to the authors own work.

Line 61 – Change raptured to ruptured.

Line 110 – Where was isopropyl alcohol used? This is the only mention of this chemical in the manuscript.

Line 117 and 127 - What is the volume of the reaction? Concentration of reactants? Number of replicates? The synthetic methods need more detail.

Line 134 – NMR solvent? What internal standard was used? How many scans?

Line 158 - Are there supplementary materials available?

Responses to Minor Points: Main text has been now amended accordingly by raised points. Addressed answers are given below.

Line 29-30 - Additional references have been now provided.

Line 52/53 – The sentence has been now changed.

Line 55 – Additional references have been now provided.

Line 58-59 - Selfcitations were unintentional since our research group is one of the leading groups in the related field. Now several references have been included in the paper to reduce the percentage.

Line 61 – Mentioned typo has been now corrected.

Line 110 – Well spotted by the reviewer. It has now been excluded from the main text.

Line 117 and 127 – No additional chemical or synthetic methods were used. All the chemicals used to conduct the reaction were reported as mols in the Materials and Methods section and also concentrations were reported in the table footnotes.

Line 134 – Details of NMR analysis has been now provided in instrumentation part.

Line 158 – Supplementary materials have now been generated and adressed.

Reviewer 2 Report

In this manuscript, the authors reported an investigation of photoinitiated cationic ring-opening polymerization of octamethylcyclotetrasiloxane (D4). The polymerization was initiated by diphenyl iodonium hexafluoophosphate (DPI). Both direct and indirect initiating systems acting at broad wave-length using benzophenone and pyrene were investigated. This is an interesting work and can be considered by this journal. This reviewer only has the following comments:

1.      In the section of Introduction, the authors should strengthen the review of literature on the same topic;

2.      There are many typos.

1), the spelling of “D4” in the title of manuscript is incorrect;

2) “higher, reaction rate”

3)  …..

Author Response

Response to Reviewer 2 Comments

General Statement: In this manuscript, the authors reported an investigation of photoinitiated cationic ring-opening polymerization of octamethylcyclotetrasiloxane (D4). The polymerization was initiated by diphenyl iodonium hexafluoophosphate (DPI). Both direct and indirect initiating systems acting at broad wave-length using benzophenone and pyrene were investigated. This is an interesting work and can be considered by this journal. This reviewer only has the following comments:

General Response: We would like to thank the Reviewer 2 for her/his kind comments and for finding our work interesting and worthy to be published on Molecules Journal.

Point 1: In the section of Introduction, the authors should strengthen the review of literature on the same topic;  

Response 1: The introduction section has now been modified and more examples from the literature have been included to strengthen the points.

Point 2: There are many typos.

1), the spelling of “D4” in the title of manuscript is incorrect;

2) “higher, reaction rate”

3)  …..

Response 2: Now, the indicated errors have been corrected and the text amended accordingly.

Reviewer 3 Report

This paper provides a new technique for the photoinitiated polymerization of octamethylcyclotetrasiloxane (D4) using DPI salts at ambient temperature. This method allows the photoinduced synthesis of polysiloxanes using common compounds at desired wavelengths under environmentally friendly and lower energy conditions, overcoming the problems associated with conventional methods and having important implications for the industrial manufacture of polysiloxanes.

It is a topic of interest to the researchers in the related areas but the paper needs some improvements before acceptance for publication. My detailed comments are as follows:

Regarding the NMR hydrogen spectra of PDMS and D4 (Fig1b), is the author sure that the chemical shift is shifted from 1.11 ppm to 0.08 ppm? It is possible that the shift is from 0.11 ppm to 0.08 ppm, please check the expression in the text.

Author Response

Response to Reviewer 3 Comments

General Statement: This paper provides a new technique for the photoinitiated polymerization of octamethylcyclotetrasiloxane (D4) using DPI salts at ambient temperature. This method allows the photoinduced synthesis of polysiloxanes using common compounds at desired wavelengths under environmentally friendly and lower energy conditions, overcoming the problems associated with conventional methods and having important implications for the industrial manufacture of polysiloxanes.

It is a topic of interest to the researchers in the related areas but the paper needs some improvements before acceptance for publication. My detailed comments are as follows:

General Response: We would like to thank the Reviewer 3 for her/his kind comments and for outlining the works potential over conventional methods.

Point 1: Regarding the NMR hydrogen spectra of PDMS and D4 (Fig1b), is the author sure that the chemical shift is shifted from 1.11 ppm to 0.08 ppm? It is possible that the shift is from 0.11 ppm to 0.08 ppm, please check the expression in the text.

Response 1: The mentioned error has now been corrected and the text amended accordingly.

Round 2

Reviewer 1 Report

The corrected manuscript “Photoinitiated cationic ring-opening polymerization of octamethylcyclotetrasiloxane” is much improved. I appreciate the corrections and suggestions that were included by the authors. However, minor issues remain.

Figure 1 – 29Si NMR spectra are typically displayed with positive to negative chemical shifts presented from left to right. This should be corrected. The experimental section indicates that NMR spectra were acquired in the solution state with a TMS internal standards. However, for this 29Si NMR, there is no resonance for TMS at 0.0 ppm. There is also no evidence in the provided 1H NMR spectrum that TMS was used as an internal standard.

Additionally, the large 29Si resonance usually associated with the glass NMR tube, found from approximately -90 to -120 ppm, is absent. Were Teflon tubes used to acquire 29Si NMR spectra? Was there post-acquisition spectral processing performed? These issues need to be clarified and included in the experimental section. 

The figure caption leads the reader to believe that the 29Si NMR spectrum for D4 is also presented. This is not the case and should be corrected.

Figure 2 – The new fitted lines a more closely resemble the experimentally acquired data and are a better guide for the eye.

Line 52 – Change electrophile enough to electrophilic enough

Line 61 – Punctuation is missing after the [55-59] references.

Line 69 – The sentence should read: These cations are capable

Line 72/73 – The newly included 29Si NMR data should be mentioned in the main body of the text.

Line 149 – The 1H NMR acquisition parameters are provided. The acquisition for parameters for 29Si NMR spectra should also be included.

Line 202 – Change polydimethoxysilane to polydimethylsiloxane.

Line 206 – fourrier-transform must be corrected to Fourier-transform.

Line 208 – Change silisium to silicon.

Author Response

Response to Reviewer 1 Comments

General Statement: The corrected manuscript “Photoinitiated cationic ring-opening polymerization of octamethylcyclotetrasiloxane” is much improved. I appreciate the corrections and suggestions that were included by the authors. However, minor issues remain.

General Response: We thank the Reviewer 1 for her/his effort and for appreciating the improvement.

Point 1: Figure 1 – 29Si NMR spectra are typically displayed with positive to negative chemical shifts presented from left to right. This should be corrected. The experimental section indicates that NMR spectra were acquired in the solution state with a TMS internal standards. However, for this 29Si NMR, there is no resonance for TMS at 0.0 ppm. There is also no evidence in the provided 1H NMR spectrum that TMS was used as an internal standard.

Response 1: Spectra is modified as advised. The used CDCl3 for 29Si-NMR was not containing TMS as a standard. Proper explanations have now been included.

Point 2: Additionally, the large 29Si resonance usually associated with the glass NMR tube, found from approximately -90 to -120 ppm, is absent. Were Teflon tubes used to acquire 29Si NMR spectra? Was there post-acquisition spectral processing performed? These issues need to be clarified and included in the experimental section.

Response 2: The resonance of the glass is visible in the mentioned region. Zoomed spectrum has now been provided in Figure 1C.

Point 3: The figure caption leads the reader to believe that the 29Si NMR spectrum for D4 is also presented. This is not the case and should be corrected.

Response 3: Well spotted. Now the caption is amended accordingly.

Point 4: Figure 2 – The new fitted lines a more closely resemble the experimentally acquired data and are a better guide for the eye.

 Response 4: We appreciate the comment of Reviewer 1 about the improvement.

Minor Points:

Line 52 – Change electrophile enough to electrophilic enough

Line 61 – Punctuation is missing after the [55-59] references.

Line 69 – The sentence should read: These cations are capable…

Line 72/73 – The newly included 29Si NMR data should be mentioned in the main body of the text.

Line 149 – The 1H NMR acquisition parameters are provided. The acquisition for parameters for 29Si NMR spectra should also be included.

Line 202 – Change polydimethoxysilane to polydimethylsiloxane.

Line 206 – fourrier-transform must be corrected to Fourier-transform.

Line 208 – Change silisium to silicon.

Responses to Minor Points: We thank Reviewer 1 for her/his attention. Main text has been now amended accordingly by raised points.
